# Exosomal MicroRNAs Released by Activated Astrocytes as Potential Neuroinflammatory Biomarkers

**DOI:** 10.3390/ijms21072312

**Published:** 2020-03-27

**Authors:** Manoshi Gayen, Manish Bhomia, Nagaraja Balakathiresan, Barbara Knollmann-Ritschel

**Affiliations:** 1Department of Neuroscience, University of Connecticut Health Center, 263 Farmington Avenue, Farmington, CT 06030, USA; gayen@uchc.edu; 2Department of Pathology, Uniformed Services University, Bethesda, MD 20814, USA; nagaraja.balakathiresan.ctr@usuhs.edu (N.B.); barbara.knollmann-Ritschel@usuhs.edu (B.K.-R.)

**Keywords:** MicroRNA, Astrocytes, neuroinflammation, Biomarkers

## Abstract

Neuroinflammation is a hallmark of several neurodegenerative diseases and disorders, including traumatic brain injury (TBI). Neuroinflammation results in the activation of glial cells which exacerbates the neuroinflammatory process by secretion of pro-inflammatory cytokines and results in disruption of glial transmission networks. The glial cells, including astrocytes, play a critical role in the maintenance of homeostasis in the brain. Activated astrocytes release several factors as part of the inflammatory process including cytokines, proteins, and microRNAs (miRNAs). MiRNAs are noncoding RNA molecules involved in normal physiological processes and disease pathogenesis. MiRNAs have been implicated as important cell signaling molecules, and they are potential diagnostic biomarkers and therapeutic targets for various diseases, including neurological disorders. Exosomal miRNAs released by astrocytic response to neuroinflammation is not yet studied. In this study, primary human astrocytes were activated by IL-1β stimulation and we examined astrocytic exosomal miRNA cargo released in a neuroinflammatory stress model. Results indicate that acute neuroinflammation and oxidative stress induced by IL-1β generates the release of a specific subset of miRNAs via exosomes, which may have a potential role in regulating the inflammatory response. Additionally, these miRNAs may serve as potential biomarkers of neuroinflammation associated with neurological disorders and injuries.

## 1. Introduction

Neuroinflammation is one of the major determinants of the progression and outcome of various neurological diseases and central nervous system (CNS) injuries, such as Alzheimer’s disease, Parkinson’s disease, Multiple Sclerosis, and traumatic brain injury (TBI). Glial cells, such as microglia and astrocytes, act as key mediators of neuroinflammation by secreting inflammatory cytokines [1]. Astrocytes are the most abundant glial cell in the brain and play a critical role in the maintenance of CNS architecture and homeostasis [2]. They regulate calcium signaling, synaptic remodeling, and the maintenance of the blood-brain barrier (BBB). In the BBB, astrocytes are an integral part of the functional barrier of the CNS neural parenchyma that restricts the access of peripheral immune cells from non-neural tissue. In the event of an injury, this functional barrier is disrupted and the astrocytes at the injury site are the key mediators of neuroinflammation, which can result in neuronal damage. Reactive astrocytes can either exert potentially beneficial effects, such as neuronal protection and BBB repair or harmful consequences such as inhibition of neuronal repair and further disintegration of the BBB. The balance between the pro- and anti-inflammatory astrocytic response to CNS insult is directed by the signaling inputs from neighboring cells and the microenvironment [3]. Among the proinflammatory cytokines, IL-1β is primarily released from microglia and is one of the potent activators of resting astrocytes. IL-1β stimulation of astrocytes not only exerts a strong pro-inflammatory signaling cascade but also drives the expression of various neuronal and glial growth-promoting genes [4].

MicroRNAs (miRNAs) are a family of small non-coding RNA molecules that act as post-transcriptional regulators of gene expression. Emerging evidence emphasizes the key role of miRNAs in regulating neuroinflammation, as is commonly observed in neurodegenerative diseases and CNS injuries [5]. MiRNAs are considered as potential biomarkers of many neurodegenerative conditions because of their stability in various body fluids, resistance to degradation after repeated freeze-thaw cycles, and ease of detection by standard PCR assays [6]. 

Exosomes are a type of extracellular vesicle which are formed by the fusion of multivesicular bodies with the plasma membrane and subsequently released from the cells. Exosomes are approximately 30–100 nm and express specific cell surface markers such as Alix and CD-63. Exosomes have a density of 1.13–1.19 g/mL in sucrose and can be sedimented at 100,000g [7,8]. Many reports have shown that miRNAs are released from the cells in the extracellular environment through exosomes. Recent studies have shown that exosomal cargo of cellular proteins, lipids, and miRNAs act as mediators of intercellular crosstalk between the effector and recipient cells [9]. Their ability to protect cargo contents from degradation and traverse across the intact BBB to reach the circulation highlights their importance as potential disease biomarkers [10]. Exosomes have also been implicated in neuroinflammatory stress responses, and neurodegenerative diseases such as Parkinson’s disease, Schizophrenia, and Alzheimer’s disease [11]. Most exosomal studies have emphasized their protein cargo, while the pathological role of the encapsulated miRNAs remains to be explored in detail. Given the role of astrocytes as key mediators of neuroinflammation, the astrocyte-specific exosomal miRNA content might play a role in determining the fate of downstream neurological complications. Few studies have reported miRNA transcriptome analysis of astrocytes from adult and human fetal brains. These studies have identified a subset of astrocyte-specific miRNAs under the normal physiological resting stage as well as IL-1β induced neuroinflammatory stress [12,13]. In this study, our goal was to identify exosomal miRNAs secreted by human astrocytes under proinflammatory and oxidative stress conditions and compare them to exosomal miRNAs released from resting astrocytes. In addition, we aim to identify the potential role of miRNAs in cellular communication, induction of inflammation, and their possible use as a neuroinflammatory biomarker.

## 2. Results

Human fetal astrocytes stimulated with recombinant human IL-1β become activated: The human fetal astrocytes treated with recombinant human IL-1β at 10 ng/mL for 24 h were evaluated for their activation by observing morphological and molecular changes. IL-1β treatment transformed the cellular morphology from flat spread out cell bodies with short processes to compact cell bodies with elongated thin processes (Figure 1A). Astrocytes treated with IL-1β had significantly increased glial fibrillary acidic protein (GFAP) immunoreactivity compared to untreated cells (Figure 1B). In addition, GFAP immunofluorescence assay with anti-GFAP antibodies showed that these are a pure population of human primary astrocytes (Figure 1C). These results suggest that treatment with IL-1β induced activation of the human fetal astrocytes.

Increased exosomal release from human fetal astrocytes following IL-1β stimulation: Exosomes were isolated from the cell supernatant of astrocytes 24 h after IL-1β stimulation by filtration and a series of ultracentrifugation steps. The presence of exosomes was confirmed by established exosomal marker Alix. Exosomes from IL-1β stimulated astrocytes had increased expression of Alix on a western blot compared with resting astrocytes which suggests that IL-1β treatment increases exosome secretion from primary astrocytes. (Figure 2). The media control group which consisted of media and exosome depleted serum showed a faint band for Alix marker indicating the presence of a small amount of exosomes in this group. This suggests that the exosome depleted serum used in the media may have some residual exosomes.

Exosomal miRNA expression from human fetal astrocytes stimulated by IL-1β: Exosomal miRNA profiling was performed with the total exosomal RNA isolated from media alone, media from resting astrocytes, and media from activated astrocytes. Due to the low concentration of exosomes in the cell culture media, the overall yield of exosomal miRNA is very low. To overcome the limitation of low exosomal miRNA content, a pre-amplification step was performed to the cDNA before the quantitative PCR of miRNA expression profiling. MiRNA expression profiling was performed using the TaqMan low-density array platform and analyzed using the StatMiner software suite. Hierarchical clustering was performed based on the overall expression of the miRNAs, which showed that the miRNA expression of IL-1β treated astrocytes was distinct from resting astrocytes and exosome free media (Figure 3). MiRNA expression from the astrocyte media and media alone groups cluster together, suggesting a low concentration of exosomal release from the resting astrocytes in media. Normalization of the miRNA expression data was performed using the global normalization method because there are no reliable or reported endogenous controls for exosomal miRNAs. Normalization of miRNA expression data resulted in two specific subsets of miRNAs: (a) miRNAs exclusively released after IL-1β treatment only in activated astrocytes, and (b) increase in miRNA concentration post-IL-1β treatment in activated astrocytes when compared to resting astrocytes. miRNAs belonging to both these subsets are presented in Table 1 and Appendix A. Five miRNAs (let-7d, miR-126, miR-130b, miR-139-5p, and miR-141-3p) were found to be secreted exclusively by IL-1β treated astrocytes. Twenty-two miRNAs were differentially expressed in the exosomes of activated astrocytes compared to resting astrocytes. Among the differentially expressed miRNAs, miR-30d and miR-195 were previously reported as biomarkers of human TBI [14]. Twenty-six miRNAs were found to be specific to resting astrocytes when compared to the media group.

Correlation and validation of miRNA expression in exosomes and astrocyte lysate using droplet digital PCR (DDPCR): Exosomal miRNA profiling showed that 5 miRNAs are exclusively secreted and 22 miRNAs are significantly upregulated by IL-1β treatment of astrocytes. Validation of selected miRNAs was performed using DDPCR. DDPCR does not require an endogenous control and provides an absolute concentration of miRNAs in the samples and hence is a preferred method for miRNA expression studies. For this study, we selected two miRNAs for validation miR-141-3p and miR-30d. MiR-141-3p was selected because of its relatively high abundance and was exclusively secreted under IL-1β stress. Moreover, increased expression of miR-141-3p has been reported after neuroinflammatory conditions such as stroke [15]. MiR-30d was selected from the pool of miRNAs due to its increased expression after IL-1β treatment. In addition, miR-30d has been reported to be enriched in astrocytes [14]. Our validation with DDPCR showed that miR-141-3p and miR-30d were significantly upregulated in the exosomes from IL-1β treated groups. Expression of both miR-141-3p and miR-30d was either not detected or present at very low concentrations in exosomes isolated from resting astrocytes and “media” controls (Figure 4A). Overall, our results from the validation experiments corroborated with the miRNA profiling data.

Further, we evaluated whether the increased concentration of exosomal RNA can be correlated to cellular miRNA expression within the astrocytes. To study this, we harvested cellular lysate from activated and resting astrocytes and performed DDPCR for miR-141-3p and miR-30d. Although the cellular RNA yield was higher compared to exosomes, cellular RNA was pre-amplified to eliminate any experimental bias validation. The cellular expression level of both miR-141-3p and miR-30d reflected the significant upregulation under IL-1β induced stress when compared to untreated astrocytes (Figure 4B). These findings suggest that the secretion of some of the miRNAs induced by IL-1β stimulation corroborated with their increased cellular expression.

IPA analysis: A total of 27 differentially expressed miRNAs in IL-1β treated astrocytes were analyzed for their potential roles during neuronal injury using IPA. We used the miRNA target filter in IPA, which uses both experimentally validated and computationally predicted mRNA targets from TargetScan, TarBase, miRecords, and the Ingenuity Knowledge Base. This filter identified 3465 experimentally validated and computationally predicted target targets for the 27 miRNAs which were identified in the IL-1β treated astrocytes. These putative mRNA targets were used for functional pathway and network analyses. Our analysis indicated that these mRNAs are possibly involved in the regulation of cell survival or cell death, suggesting that miRNAs altered by IL-1β treatment may lead to alteration of pathways leading to cell death (Figure 5A). The other putative roles of these miRNAs are listed in Appendix A. A network map of miRNA-mRNA interaction was constructed using experimentally validated as well as computationally predicted which shows their possible role in apoptosis, neuroinflammatory, and neurodegeneration signaling pathways (Figure 5B). MiRNAs are predicted to target several molecules which are key mediators in inflammatory and apoptotic pathways. The mRNAs for *BCL-2* and *TLR-4* are predicted to be targeted by more than 1 miRNAs. The mRNAs for *BCL2L1, BAX,* and *CASP3* are also predicted to be targeted by the dysregulated miRNAs. These miRNAs are known to play a key role in the apoptotic pathway. The mRNAs for *BACE* and *APP* are also shown to be targeted by the miR-29a and miR-520d-3p which show a possible role of the miRNAs in the regulation of β-amyloid protein. In summary, this analysis suggests that exosomal miRNAs have a possible involvement in downstream apoptotic and neuroinflammatory cascade by targeting intermediate mRNA molecules.

## 3. Discussion

The objective of this study was to evaluate the release of exosomal miRNAs by activated astrocytes under inflammatory stress. Inflammation is a key feature of many neurodegenerative diseases and disorders. Exosomal miRNAs released from activated astrocytes were studied to identify possible biomarkers which can be used as a liquid biopsy for neuroinflammation. Neuroinflammation is a complex process that can be initiated in the CNS by a variety of mechanisms such as pathogen detection, neuronal injury, or BBB damage. Cytokines play a key role in initiating and maintaining neuroinflammation. Therefore, for this study, we used IL-1β to induce an inflammatory response in the cultured astrocytes. IL-1β was chosen because it is a primary cytokine released by activated microglia at the site of brain injury and is a prominent contributor to inflammatory signaling cascades. Previous studies have shown that cellular and/or exosomal miRNAs of mouse astrocytes are dysregulated in response to inflammatory stress [13,16]. However, there are no reports on the release of astrocytic exosomal miRNAs from activated human astrocytes. In this study, we used primary human astrocytes to study the miRNAs released in exosomes under inflammatory stress.

Exosome free serum for cell culture was used to limit the introduction of exogenous bovine exosomes present in serum, which is known to be a rich source of exosomes. However, we detected a small quantity of exosomes in the media alone. This may be due to the technical limitations of the exosomal depletion method used by the vendor. However, the presence of this small amount of exosomes is not significant in comparison to the exosomes isolated from the astrocytes and is unlikely to be altered by treatment with IL-1β. Our data from exosomal miRNA profiling from activated astrocytes screening identified a very distinct miRNA expression profile at 24 hr post-IL-1β induced inflammatory stress. Following the treatment with IL-1β, we observed an increased secretion of exosomes by astrocytes and the miRNA cargo in the exosomes. 

Further, clustering analysis concurred with the earlier observation that the exosomal miRNAs from the IL-1β treated group clustered together, suggesting a distinct miRNA expression profile in comparison to the untreated cells. For normalization of the data, we adopted a global normalization approach as there is no well-characterized exosomal miRNA endogenous control. Global normalization results showed two distinct miRNA subsets. The first subset was exclusively expressed in the exosomes from cells treated with IL-1β, while we did not observe any significant expression in the untreated group. In the second subset, we identified miRNAs that were significantly upregulated by several folds in the exosomes from cells treated with IL-1β. Many of these miRNAs, which we identified in the exosomal miRNAs after IL-1β treatment, have been reported to be present in the blood of several neurodegenerative conditions. Our laboratory has previously reported altered expression of miR-30d and miR-195 in human TBI [17]. MiR-30d has also been reported as a severity indicator of systemic inflammation. MiR-30d is known to upregulate pro-inflammatory cytokines including IL-1β via foxo3a regulation and regulates neural autophagy and apoptosis [18]. In addition, it has also been reported as a potential blood-based biomarker of Alzheimer’s disease [19]. We noted miR-141-3p as the most abundant exosomal miRNA exclusively present in the inflammatory stress (Table 1).

Further validation showed miR-141-3p to have significantly upregulated expression in astrocytes as well as within exosomes. MiR-141-3p has been reported as a serum plasma biomarker of Alzheimer’s disease, a regulator of glioma metastasis, and chronic inflammatory pain [20,21,22]. MiR-141-3p levels were reported to be decreased in brain tissues in prion disease [23] and involved in CXCL12β mediated leukocyte migration to the site of inflammation [24]. Hence, blood miRNA expression following brain trauma and other neuroinflammatory biomarkers could be due to an astrocyte-mediated neuroinflammatory response.

In order to identify the potential pathways modulated by the miRNAs, IPA analysis was performed. A total of 20 miRNAs were found associated with neuroinflammatory pathways. Amongst them, miR-let-7d, miR-29a, miR-30d, miR-31-3p, miR-93-3p, and miR-145-5p have previous experimental validation for targeting proteins such as GFAP, aquaporin, vimentin, and amyloid precursor protein, respectively [2,25,26]. Previously published reports by others have implied the role of these proteins in different neurological disorders and brain trauma [24,25,26,27,28]. IPA further underscored the significance that miRNAs secreted by astrocytes via exosomes could be potential mediators of neuroinflammatory signaling cascades. Additional studies to explore the effect of activated astrocyte secreted-specific miRNAs in a neuron-glia co-culture model or brain slice culture will be required to elucidate more specific signaling pathways.

## 4. Methods

Primary astrocyte culture and IL-1β treatment: Human fetal astrocytes were commercially acquired from Thermo Fisher Scientific Inc (Catalog# K1884). The cells were seeded at the recommended concentration of 4 × 10^4^ cells/ cm^2^ in a Geltrex matrix coated T25 tissue culture flask. The tissue culture flasks were treated with Geltrex matrix (Thermo Fisher Scientific, Waltham, WI, USA) according to the manufacturer’s protocol. The astrocytes were grown in complete astrocyte medium as recommended by the manufacturer’s protocol. The cell media was replaced with heat inactivated exosome-depleted fetal bovine serum media supplement (SystemBio, Palo Alto, CA, USA) 24 hr prior to IL-1β stimulation. Human recombinant IL-1β (Thermo Fisher Scientific) was used for stimulating the astrocytes at the concentration of 10 ng/ml for 24 hr. The cell supernatant was collected at the end of IL-1β treatment. The experiment was performed in biological triplicates.

Exosome isolation: The astrocytes were treated with IL-1β for 24 hr prior to cell supernatant collection. Control supernatant was collected from untreated astrocytes. Exosomes were harvested from the cell supernatant as described previously [14] by the ultracentrifugation method. Briefly, the supernatant was centrifuged at 400× *g* at 4 °C for 15 min to remove cellular debris. The supernatant was clarified with a 0.22 µm filter to remove apoptotic bodies, cellular debris and microvesicular bodies which generally have a size greater than 200 nm. The clarified supernatant was centrifuged at 100,000× *g* at 4 °C for 90 min (Beckman Coulter. Rotor type SW40Ti, Brea, CA, USA) to isolate crude exosomal pellet. The exosomal pellets were resuspended in phosphate buffer saline (PBS) (Thermo Fisher Scientific) and for subsequent centrifugation at 100,000× *g* at 4 °C for 90 min (Beckman Coulter. Rotor type SW40Ti). The pellet obtained was resuspended in phosphate buffer saline (PBS) and stored at -80 °C for further use. As a negative control, exosome isolation using ultracentrifugation was also performed with an equal volume of cell culture media supplemented with exosome depleted serum (System Bio Inc).

Western blotting: The presence of exosomes in the pellet was confirmed by western blot analysis using antibodies for Alix and CD63, which are reported markers for exosomes. Seven µg of total protein from exosome pellet was resolved on 4%-12% Bis-Tris gel (Thermo Fisher Scientific) followed by transfer on a nitrocellulose membrane (Bio-Rad). Membrane blocking was done with 5% nonfat dry milk (Santa Cruz Biotechnology, Inc., Dallas, TX, USA) in 1X Tris-buffered saline with tween (TBST) (Bioexpress, Inc.) for 1 h. The membrane was incubated overnight with primary antibody Alix (Santa Cruz Biotechnology) or diluted to 1:2500 in 3% non-fat dry milk. Astroglia activation was confirmed by GFAP and beta-actin western blot analysis. Cellular proteins were isolated by cell lysis using Radio immunoprecipitation assay (RIPA) lysis buffer (Thermo Fisher Scientific, Inc.) according to the manufacturer’s protocol. 50 µg of cellular proteins were resolved on 4–12% Bis-Tris gel (Thermo Fisher Scientific) for immunoblotting. Primary antibodies for GFAP (Santa Cruz Biotechnology) and beta-actin (Cell Signaling) were diluted to 1:3000 in 3% non-fat dry milk. Secondary antibodies for anti-mouse IgG labelled with Horseradish peroxidase (Cell Signaling Inc., Danvers, MA, USA) and anti-rabbit IgG HRP (Cell Signaling) were diluted to 1:3000 and 1:2500 in 3% non-fat dry milk. The immunoblots were developed using ECL plus reagent (Thermo Fisher Scientific, Inc.). 

RNA isolation and quality check: Total RNA was isolated from the exosomes and cells using the RNAqueous Total RNA Isolation kit (Thermo Fisher Scientific) and Trizol (Thermo Fisher Scientific), respectively, according to the manufacturer’s protocol. Briefly, the exosome pellet was treated with denaturing solution by addition of the acid:phenol:chloroform for phase separation. The aqueous phase was collected and 1.25 volumes of 100% ethanol were added. The solution was passed through an RNeasy MinElute spin column for binding the RNA. The RNA was eluted in nuclease-free water and stored at −80 °C for further use. The total RNA was quantified using a Nanodrop spectrophotometer. The miRNA quality was analyzed with the bioanalyzer using a small RNA kit (Agilent Inc) as per the manufacturer’s protocol. 

Exosomal miRNA profiling: For miRNA profiling, reverse transcription (RT) was performed with 6.4 ng of total exosomal RNA using the TaqMan miRNA RT kit (Thermo Fisher Scientific) and Megaplex Human primer pool A/B (v3.0) (Thermo Fisher Scientific) according to the manufacturer’s protocol. The limitation of low RNA yield from exosomes was compensated by pre-amplification of the cDNA after RT using the TaqMan PreAmp Master Mix (Thermo Fisher Scientific) and Megaplex PreAmp primers human pool A/B (v3.0) (Thermo Fisher Scientific). RT and preamp reactions were carried out on Veriti 96-Well Thermal Cycler (Thermo Fisher Scientific) according to the manufacturer’s recommended thermal cycling conditions. The real-time PCR reaction was prepared by mixing 9 μL of undiluted pre-amplification product to 450 μL of 2X TaqMan Universal PCR Master Mix, No AmpErase UNG (Thermo Fisher Scientific) and nuclease-free water to a final volume of 900 µL. Then, 100 µL of the PCR reaction mix was loaded onto each row of the 384-well TaqMan Low-Density Human MicroRNA array cards v3.0 (Thermo Fisher Scientific). The PCR reaction was carried out at default thermal-cycling conditions in AB7900 Real-Time HT machine (Thermo Fisher Scientific). The analysis of the miRNA expression profile was performed using real-time StatMiner software (Perkin Elmer Inc., Waltham, WI, USA) to identify significantly altered miRNAs. For relative quantification of miRNAs released by IL-1β stimulated astrocytes and resting astrocytes, the following steps were performed in the StatMiner software suite: quality control of biological replicates, filtering of miRNAs expression having cycle threshold (Ct) values below 35 cycles and the detection of expression in all biological replicates of calibrator and target. Statistically significant miRNAs were selected based on the following stringent parameters using Benjamin-Hochberg false discovery rate (FDR) corrections conservatively selecting data with adjusted *p*-values, and *p*-value lower than both 0.01 and 0.05. 

MiRNA expression validation using droplet digital PCR: MiRNA expression was validated using the droplet digital PCR (DDPCR) (Bio-Rad) as per the manufacturer’s recommended protocol. Briefly, 10 ng of total cellular or exosomal RNA was used to perform RT reaction using specific TaqMan MicroRNA assay primers (Thermofisher Scientific, Inc.) according to the manufacturer’s protocol. Pre-amplification of cDNA was performed to compensate for the low RNA yield. DDPCR was performed using the QX200 droplet generator (Bio-Rad., Hercules, CA, USA) as per the manufacturer’s recommended protocol. Briefly, a PCR master mix was made, which consisted of 1µL of RT product, miRNA specific 20X TaqMan probes, nuclease-free water, DDPCR super mix for probes (No dUTP) (BioRad, Inc.), and droplet generation oil for probes (BioRad, Inc., Hercules, CA, USA). Droplets were made using the droplet generator. After the oil droplet formation, the plate was sealed using the PX1 PCR plate sealer (BioRad Inc.,Hercules, CA, USA). PCR reaction was performed as per recommended thermal cycling conditions using C1000 Touch thermal cycler (BioRad Inc., Hercules, CA, USA). The concentration of miRNA per reaction volume was analyzed using the QX200 droplet reader (Bio-Rad Inc). All the reactions were performed in duplicates.

Ingenuity Pathway Analysis: Functional pathway analysis of altered miRNAs and their association with neuroinflammation related gene targets were performed using the Ingenuity Pathway Analysis (IPA) program (IPA, QIAGEN Redwood City, CA, USA). For clinical correlation analysis, data comparing changes in miRNA expression are described as means with a standard error of the mean. To analyze the differences between group means we used analysis of variance (ANOVA) after the assessing for distribution and variance. Multiple comparisons were performed using the Games–Howell test. Significance was set at 0.05.

## 5. Conclusions

This is the first study to identify the miRNA expression profile secreted by astrocytes in response to acute neuroinflammatory stress. We identified and extensively characterized miRNA cargo released by astrocytes under IL-1β stimulated acute neuroinflammatory stress. A subset of five miRNAs was found to be specifically released by the activated astrocytes and may serve as potential biomarkers of CNS inflammation. These markers can be further validated in a clinical cohort of patients with neuroinflammatory conditions. Further studies are essential for deciphering the mechanisms of exosomal miRNA mediated inflammatory signaling cascades in the CNS. 

### Data Availability Statement

The cell culture and microRNA profiling data used to support the findings presented in this study are included in the manuscript and as Appendix A.

## Figures and Tables

**Figure 1 ijms-21-02312-f001:**
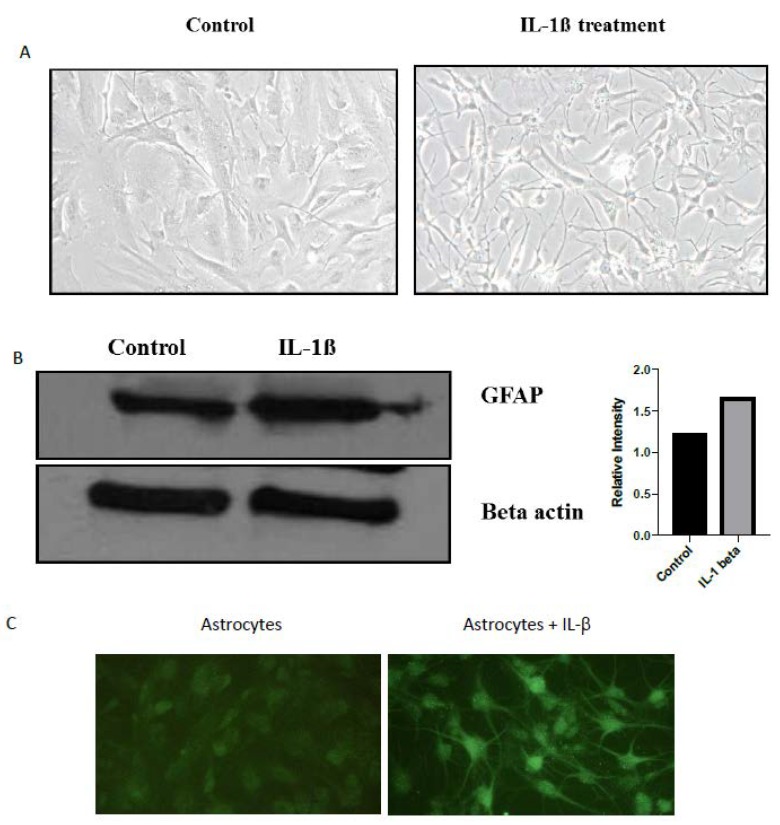
Astrocyte activation on IL-1β stimulation. (**A**) Human fetal astrocytes were treated with IL-1β at a concentration of 10 ng/mL for 24 hr. The morphology of the treated astrocytes acquired elongated processes with a compact cell body that is characteristic of activated astrocytes. (**B**) Western blot of the cellular protein shows an increase in glial fibrillary acidic protein (GFAP) on IL-1β stimulation. Relative quantitation shows an increase in GFAP concentration after treatment with IL-1β. (**C**) Immunofluorescence image showing expression of GFAP in astrocytes with and without treatment with IL-1β at 24 h post-treatment.

**Figure 2 ijms-21-02312-f002:**
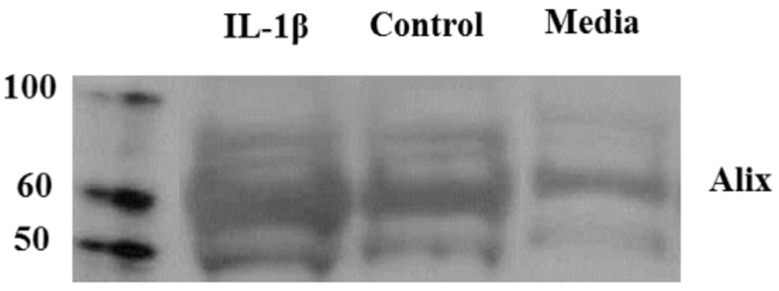
Exosome release from activated astrocytes. Western blot on the exosomes from astrocytes treated with or without IL-1β demonstrates an increase in the exosomal marker Alix.

**Figure 3 ijms-21-02312-f003:**
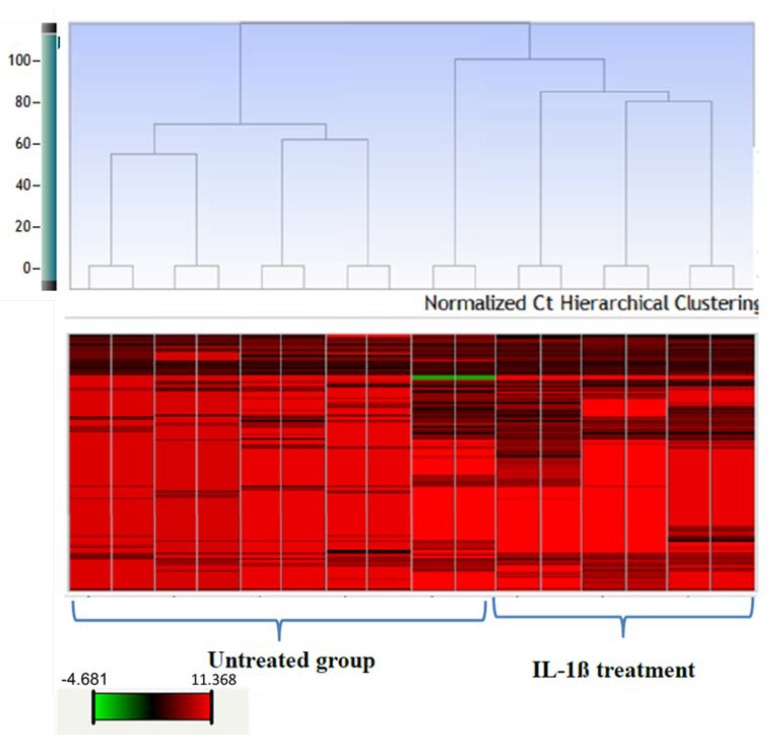
Hierarchical clustering (HC) of miRNAs from astrocytic exosomes. The dendrogram was constructed by HC using the complete linkage method together with the Pearson correlation measure based on the delta Ct values normalized using the z-score normalization method. The red and green colors in the heat map represent the miRNA expression in terms of delta Ct value. MiRNA profiling showed distinct clusters of miRNAs in IL-1β treated astrocytes and untreated astrocytes.

**Figure 4 ijms-21-02312-f004:**
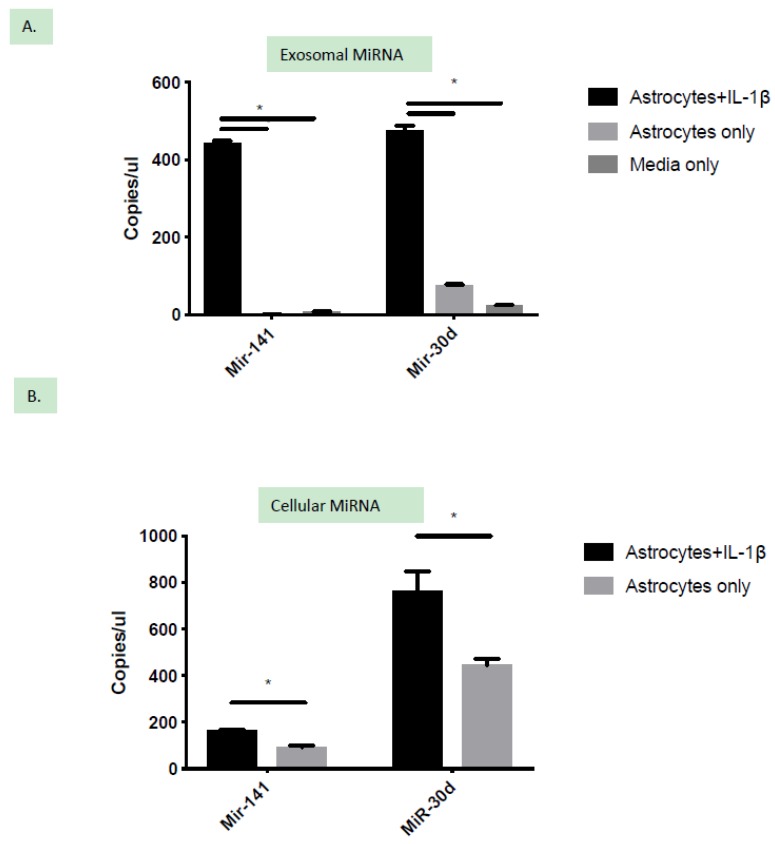
Validation of enhanced expression of miRNAs upon IL-1β treatment. (**A**) Release of miR-141-3p and miR-30d was found to be significantly increased within the exosomes of IL-1β stimulated astrocytes. (**B**) The cellular expression level of miR-141-3p and miR-30d was confirmed to be significantly increased within the activated astrocytes. * *p* < 0.05; student t-test.

**Figure 5 ijms-21-02312-f005:**
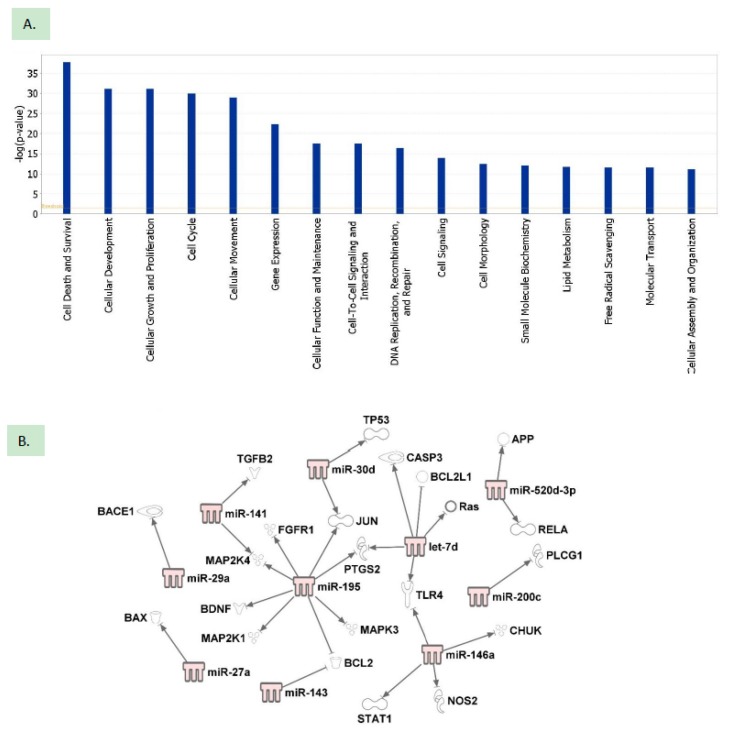
Ingenuity pathway analysis of the exosomal miRNAs dysregulated upon IL-1β treatment. (**A**) Cellular and molecular functional analysis of the miRNAs included all the determined experimental targets predicted cell death and survival as the topmost biological function of the exosomal miRNAs; (**B**) A network analysis of IL-1β treated astrocytes miRNAs and gene targets commonly implicated in apoptosis, neuroinflammatory, and neurodegeneration showing possible roles of the selected miRNAs in these pathways.

**Table 1 ijms-21-02312-t001:** MiRNA expression from exosomes secreted by primary human astrocytes following IL-1β treatment.

***MiRNAs released exclusively on IL-1β treatment***			
**S. No.**	**MiRNA**	**MIMAT ID**	***P value***
1	hsa-miR-141	MIMAT0000432	1.21E-04
2	hsa-miR-139-5p	MIMAT0000656	3.52E-02
3	hsa-miR-126	MIMAT0000445	3.77E-02
4	hsa-miR-130b	MIMAT0000691	1.92E-02
5	hsa-let-7d	MIMAT0000065	1.87E-02
***MiRNAs upregulated within exosomes on IL-1β treatment***			
**S. No.**	**MiRNA**	**MIMAT ID**	***P value***
1	hsa-miR-143	MIMAT0000435	2.20E-02
2	hsa-miR-145	MIMAT0000437	5.88E-03
3	hsa-miR-146a	MIMAT0000449	9.59E-03
4	hsa-miR-152	MIMAT0000438	2.48E-02
5	hsa-miR-192	MIMAT0000222	4.96E-02
6	hsa-miR-194	MIMAT0000460	2.49E-02
7	hsa-miR-195	MIMAT0000461	4.78E-02
8	hsa-miR-197	MIMAT0000227	1.59E-03
9	hsa-miR-200b	MIMAT0000318	2.06E-02
10	hsa-miR-200c	MIMAT0000617	2.47E-02
11	hsa-miR-203	MIMAT0000264	9.92E-03
12	hsa-miR-215	MIMAT0000272	2.41E-02
13	hsa-miR-27a	MIMAT0000084	4.25E-02
14	hsa-miR-29a	MIMAT0000086	4.59E-02
15	hsa-miR-30d	MIMAT0000245	5.00E-03
16	hsa-miR-31	MIMAT0000089	0.012534
17	hsa-miR-375	MIMAT0000728	3.26E-02
18	hsa-miR-494	MIMAT0002816	4.76E-02
19	hsa-miR-520d-3p	MIMAT0002856	3.11E-02
20	hsa-miR-539	MIMAT0003163	7.32E-04
21	hsa-miR-885-5p	MIMAT0004947	9.42E-04
22	hsa-miR-93*	MIMAT0004509	1.58E-02

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
