# Peer review of "Exosomal MicroRNAs Released by Activated Astrocytes as Potential Neuroinflammatory Biomarkers"

_ijms, 2020, doi:10.3390/ijms21072312_

Round 1
Reviewer 1 Report
The study by Gayen et al. presents changes in exosomal miRNAs as markers and potential effectors of astrocyte activation in vitro. This is a sound study, although purely descriptive in scope. It is not clear how relevant these results are in vivo or in a clinical context. Additionally, a few minor points require correction:
the profiling results should be made available on a public database Figure 3 - the color scale of the heat map is problematic, needs to be adjusted for the figure to be more informative. There are minor spelling and style errors throughout the manuscript; the symbol for the Greek letter "beta" seems incorrect throughout
Author Response
Response to Reviewers Comments
We thank the reviewer for their review of the manuscript and constructive criticism. We have addressed the issues raised by the reviewers and revised the manuscript accordingly in track changes. A point by point response to each comment is provided below.
Reviewer 1
Comment 1: The study by Gayen et al. presents changes in exosomal miRNAs as markers and potential effectors of astrocyte activation in vitro. This is a sound study, although purely descriptive in scope. It is not clear how relevant these results are in vivo or in a clinical context.
Response 1: The main goal of this study was to understand the miRNAs secreted from activated astrocytes. Astrocyte activation or inflammation is central at the onset of many neurodegenerative diseases, however early detection of these diseases if often difficult clinically. Blood based biomarkers can be useful to detect these disorders at an early stage. The results of our study provide initial evidence that activated astrocytes release exosomal miRNAs which could be used for the detection of the onset of neurodegeneration. Although, the results presented here are based on an in vitro study, these results can be used for validation of a clinical cohort of patients with neurodegenerative conditions. We modified the conclusion section of the manuscript accordingly to highlight the relevance of this study for a clinical setting.
Comment 2: A few minor points require correction: the profiling results should be made available on a public database Figure 3 - the color scale of the heat map is problematic, needs to be adjusted for the figure to be more informative. There are minor spelling and style errors throughout the manuscript; the symbol for the Greek letter "beta" seems incorrect throughout.
Response 2: We agree with the reviewer regarding making the profiling results publically available. As per the IJMS policy, we have chosen the option to uploaded the profiling results as a supplementary data in a Microsoft excel sheet.
The color scale indicating the corresponding delta Ct values in the heat map is now included in the revised figure as per reviewers’ comment. We used a licensed Statminer Software to perform the analysis and generate this figure. However, the license for this software is expired and it is not available for repurchase. Therefore, we cannot change the color scheme, however including the color scale should provide sufficient information for the readers to interpret the data.
We appreciate reviewer’s comments and have revised the manuscript for its spelling and style errors. The manuscript has been spell-checked again. The symbol for the Greek letter beta is corrected throughout the manuscript.
Reviewer 2 Report
The manuscript covers an interesting topic and is focused on a simple and reproducible astrocytes cell model in order to investigate miRNA content in extracellular vesicles following activation. However, some sentences are unclear, other need further details, otherwise it is difficult to understand whether data were correctly analysed (see Major points). In the Introduction, the use of the term exosomes should be clearly explained, because it indicates a clear biosynthetic origin from Multivesicular bodies, but from an experimental point of view exosomes are very difficult to separate from membrane microvesicles (see Kowal et al., PNAS 2016).
Major points
Some data are difficult to interpret according to the technical description. Figure 2 comment is that exosomes release from IL1b treated samples show an increase of Alix marker. However, there is no mention about the quantity of exosomes that was loaded in each lane and, most relevant, how the sample named “Media” was prepared. Besides, the validation of CD9 marker is mentioned in Materials and Methods section, but no data on CD9 are shown.
In the results section, authors claim that “MiRNA exosomal profiling was performed with the total exosomal RNA isolated from media alone, media from resting astrocytes, and media from activated astrocytes.”. However, in the Materials and Methods there is no explanation on how extracellular vesicles were isolated from media alone. In addition, there are no details on exosome depletion from serum, although in the Results section it is stated that “Although the cells were maintained in "Exosome free serum" for the duration of the IL-1ß stimulation to avoid contamination with any exogenous exosomes, some residual exosomes were detected in the media.” Therefore, it is not possible to understand whether exosomal miRNA were isolated from this "Exosome free serum", and what do authors mean for “some residual exosomes were detected in the media”. It is not possible to write “some”. How were these “residual” exosomes evaluated? Howe were these culture medium exosomes depleted? Another methodological concern is related to the pre-amplification step: how could authors compare expression level after a pre-amplification step, that can easily introduce quantitative bias?
miRNA have been shown to be released extracellularly either associated or not associated to extracellular vesicles. Therefore, a pre-digestion step with RNAse is usually carried out in order to ensure to amplify only miRNA protected by a vesicular structure: why did not authors carry out a RNase digestion step?
Other points:
In the protocol from exosomes separation, authors did not use serial centrifugation steps: could authors comment on this choice? Authors also state that “The supernatant was clarified with a 0.22 μm filter to remove microvesicular bodies.”. This step should be clearly explained to the readers
Authors carried out GFAP staining: do they carry out of other markers, to test possible differentiation in other neural cell types in their culture conditions?
In the Abstract, the sentence “The role of secretory miRNAs in the astrocytic response to neuroinflammation is not known.” is misleading, as it is not clear that the aim of the study is to investigated miRNA associated with exosomes
In the Introduction, the sentence “It has been shown that miRNAs are released from cells into the extracellular vesicles called exosomes, which are of approximately 30-200 nm in size.” is unclear/uncorrect. Why do authors write “extracellular vesicles called exosomes”? Do authors want to underlie that miRNA cannot be released in other types of vesicles? Could authors define for the readers what exosomes are among the family of extracellular vesicles?
In the Introduction , the sentence “Their ability to protect cargo contents from degradation and traverse across the intact BBB highlight their importance as potential disease biomarkers [8].” is also not clear: do authors refers to the ability of circulating EVs to cross the BBB or neural EVs to reach the circulation?
IPA network should be illustrated in more detail, adding at least a few sentences
Author Response
Response to Reviewers Comments
We thank the reviewer for their review of the manuscript and constructive criticism. We have addressed the issues raised by the reviewers and revised the manuscript accordingly in track changes. A point by point response to each comment is provided below.
Reviewer 2
Comment 1: Figure 2 comment is that exosomes release from IL1b treated samples show an increase of Alix marker. However, there is no mention about the quantity of exosomes that was loaded in each lane and, most relevant, how the sample named “Media” was prepared. Besides, the validation of CD9 marker is mentioned in Materials and Methods section, but no data on CD9 are shown.
Response 1: 7 μg of total exosomal protein was loaded in each lane of the protein gel. “Media” sample consisted of exosomes isolated from media alone with exosome free serum. We have modified the methods sections to include these details. CD9 marker was mentioned by mistake in the methods section and has been removed.
Comment 2: In the results section, authors claim that “MiRNA exosomal profiling was performed with the total exosomal RNA isolated from media alone, media from resting astrocytes, and media from activated astrocytes.”. However, in the Materials and Methods there is no explanation on how extracellular vesicles were isolated from media alone.
Response 2: We have edited the “exosome isolation” section in methods for clarity regarding the steps of exosome isolation. Exosomes were isolated in a similar manner from an equal volume of media. This is clarified in the methods section.
Comment 3: In addition, there are no details on exosome depletion from serum, although in the Results section it is stated that “Although the cells were maintained in "Exosome free serum" for the duration of the IL-1ß stimulation to avoid contamination with any exogenous exosomes, some residual exosomes were detected in the media.” Therefore, it is not possible to understand whether exosomal miRNA were isolated from this "Exosome free serum", and what do authors mean for “some residual exosomes were detected in the media”. It is not possible to write “some”. How were these “residual” exosomes evaluated? Howe were these culture medium exosomes depleted? Another methodological concern is related to the pre-amplification step: how could authors compare expression level after a pre-amplification step, that can easily introduce quantitative bias?
Response 3: Cell culture experiments performed in this study used exosome depleted serum. The primary reason for that was to avoid any contamination from bovine exosomes. The exosome depleted serum was commercially purchased from System Bio Inc. However, when we performed the western blot for the exosome specific marker Alix, a faint band for Alix was detected in the “media” control. This suggests that the exosome depleted serum was not completely free of exosomes. This is due to technical limitations of exosome depletion which leads to the presence of a small number of exosomes. We have revised the methods and results section accordingly for clarity.
The reviewer has also raised an important point of pre-amplification bias in real-time PCR assays. We have performed pre-amplification due to the very low amount of RNA recovery from the exosomes. The kits which we used have been shown to have a very low pre-amplification bias as per manufacturer data. We still wanted to verify the accuracy of our data and hence performed individual droplet digital PCR (DDPCR) validation for two miRNAs for which the data is presented in figure 4. DDPCR was done without the pre-amplification step, so we were able to validate the results of the profiling. Moreover, we have successfully performed and validated our profiling results with the same method previously (Bhomia et al 2016, Balakathiresan et al l 2014).
Comment 4: miRNA have been shown to be released extracellularly either associated or not associated to extracellular vesicles. Therefore, a pre-digestion step with RNAse is usually carried out in order to ensure to amplify only miRNA protected by a vesicular structure: why did not authors carry out a RNase digestion step?
Response 4: We agree with the reviewer that miRNAs can be release directly in the media and may contribute to overall RNA amplification and profiling results. In this study, we isolated exosomes using ultracentrifugation which leads to an exosomal pellet. Therefore, there is no contamination of the extracellular RNA since only a pure fraction of exosomal pellet was used for RNA isolation.
Comment 5: In the protocol from exosomes separation, authors did not use serial centrifugation steps: could authors comment on this choice? Authors also state that “The supernatant was clarified with a 0.22 μm filter to remove microvesicular bodies.”. This step should be clearly explained to the readers
Response 5: We have performed a single ultracentrifugation for exosome isolation based on a published manuscript which has standardized the exosome isolation from cell culture media. The primary reason for isolation of exosome using single step ultracentrifugation was to reduce the loss of exosome yield which is associated with serial ultracentrifugation. The media volume was limiting and hence we wanted to enhance the yield of exosomes for our experiments. We have made modifications in the methods section for clarity regarding the use of a 0.22 μm filter.
Comment 6: Authors carried out GFAP staining: do they carry out of other markers, to test possible differentiation in other neural cell types in their culture conditions?
Response 6: The primary astrocytes used for this study were commercially acquired from Thermofisher Scientific (Cat no K1884). These cells were tested for purity by the vendor. To ensure the purity of the culture, we performed GFAP staining and all the cells stained positive for GFAP. Additionally, morphological examination of the cells after treatment with IL-1beta showed a characteristic astrocytic process. Therefore, we concluded the cells are pure and did not test them further for any neural cell contamination.
Comment 7: In the Abstract, the sentence “The role of secretory miRNAs in the astrocytic response to neuroinflammation is not known.” is misleading, as it is not clear that the aim of the study is to investigated miRNA associated with exosomes.
Response 7: We appreciate the reviewer’s comment and have change the sentence in the abstract for better clarity and alignment with the objective of the study.
Comment 8: In the Introduction, the sentence “It has been shown that miRNAs are released from cells into the extracellular vesicles called exosomes, which are of approximately 30-200 nm in size.” is unclear/uncorrect. Why do authors write “extracellular vesicles called exosomes”? Do authors want to underlie that miRNA cannot be released in other types of vesicles? Could authors define for the readers what exosomes are among the family of extracellular vesicles?
Response 8: We have corrected the sentence and added a description about family of extracellular vesicles to accurately describe exosomes. We have mentioned that a majority of miRNA content released from the cells are present in the exosomes. We do not mean to imply that miRNAs are not released in the other extracellular vesicles and apoptotic bodies and hence clarified this in the text. We have added this in the introduction for more clarity.
Comment 9: In the Introduction , the sentence “Their ability to protect cargo contents from degradation and traverse across the intact BBB highlight their importance as potential disease biomarkers [8].” is also not clear: do authors refers to the ability of circulating EVs to cross the BBB or neural EVs to reach the circulation?
Response 9: Exosomes are known to cross blood brain barrier. The references (8) and (9) mentioned in the text refer to the ability of EVs containing amyloid beta, tau protein, and other markers of neurodegenerative diseases to cross the BBB and enter the peripheral circulation, hence, they have been used as diagnostic tools. Exosomes have also been used to deliver drugs to brain in which circulatory exosomes have been shown to cross BBB and reach brain (Dongfen Y et al 2017). However, in the context of this manuscript we have highlighted the role of neural EVs to reach the circulation. We have modified the introduction to explain this clearly.
Comment 10: IPA network should be illustrated in more detail, adding at least a few sentences.
Response 10: We agree with the reviewer and have added more details in the IPA results in the results section of the manuscript.
References
- Bhomia, M.; Balakathiresan, N.S.; Wang, K.K.; Papa, L.; Maheshwari, R.K., A Panel of Serum MiRNA Biomarkers for the Diagnosis of Severe to Mild Traumatic Brain Injury in Humans. Sci Rep 2016, 6, 28148.
- Balakathiresan, N.S, Chandran, R, Bhomia M, Jia M, Li H;Maheshwari RK. Serum and amygdala microRNA signatures of posttraumatic stress: fear correlation and biomarker potential. J Psychiatr Res, 2014, 56,65-73.
- Dongfen Y, Yuling Z , William B, Kristin M Bullock, Matthew H , Elena B , Alexander V Kabanov. Macrophage Exosomes as Natural Nanocarriers for Protein Delivery to Inflamed Brain.2017, Biomaterials , 142, 1-12
Round 2
Reviewer 2 Report
The revised version of the manuscript has improved the first version, but a few points are still confused, and some correction have introduced clear mistakes that need to be fixed. The sentence that has been added at lines 58-60 is confused and absolutely incorrect: “Extracellular vesicles are a heterogeneous population of intracellular microvesicular bodies (MVBs), apoptotic bodies, and ectosomes. Exosomes are MVBs that are approximately 30-100 nm and express specific cell surface markers such as Alix and CD-63.” Extracellular vesicles are not an “heterogeneous population of intracellular microvesicular bodies (MVBs)” and exosomes are not MVBs. Exosomes originate from the inward budding of late endosome membrane. For this reason, late endosomes become full of intraluminal vesicles and take the name of Multi Vesicular Bodies. Intraluminal Vesicles are released upon exocytosis and take the name of “exosomes”. Authors should correctly explain what exosomes are and add suitable references for the readers.
Minor points:
In the reply to Comment 1, authors write that “Media” sample consisted of exosomes isolated from media alone with exosome free serum”. What do authors mean? I suppose tat they use exosome free serum but they were still able to precipitate exosome, as stated in the reply. However, this point in the text is still unclear. At line 100, authors wrote that “Exosomes were isolated from an equal volume of the cell culture media containing heat inactivated exosome-depleted fetal bovine serum media supplement (SystemBio Inc) has beenwhich is referred to as “mMedia control”. if authors suspected/detected that exosome-free medium was not properly depleted, they should clearly explain that, as it is clearly explained in the reply.
Line 192 refers to the same point and the sentence is again unclear “Although the cells were maintained in "Exosome free serum (SystemBio)" for the duration of the IL-1βß stimulation to avoid contamination with any serum exogenous exosomes, some residual exosomes were also detected in the exosome free media. This suggests that IL-1βß mediated astrocyte activation increases exosome secretion from primary astrocytes.” Why the presence of residual exosomes in the exosome free media suggests that IL-1βß mediated astrocyte activation increases exosome secretion from primary astrocytes? The Media alone sample should be clearly explained
Line 96: Please correct the sentence “to pellet crude exosomal pellet”
Line 274. The sentence “MRNAs for BCL-2, Bax and caspase 3 are among the few molecules which are involved in inflammatory and apoptotic pathway” is not precise, as many other molecules are involved. Authors should reformulate this sentence
Author Response
Response to Reviewer Comments
Comment 1: The sentence that has been added at lines 58-60 is confused and absolutely incorrect: “Extracellular vesicles are a heterogeneous population of intracellular microvesicular bodies (MVBs), apoptotic bodies, and ectosomes. Exosomes are MVBs that are approximately 30-100 nm and express specific cell surface markers such as Alix and CD-63.” Extracellular vesicles are not an “heterogeneous population of intracellular microvesicular bodies (MVBs)” and exosomes are not MVBs. Exosomes originate from the inward budding of late endosome membrane. For this reason, late endosomes become full of intraluminal vesicles and take the name of Multi Vesicular Bodies. Intraluminal Vesicles are released upon exocytosis and take the name of “exosomes”. Authors should correctly explain what exosomes are and add suitable references for the readers.
Response: We agree with the reviewer’s comment. The sentence has been corrected with an accurate definition of the exosomes. Appropriate references are also incorporated.
Comment 2” In the reply to Comment 1, authors write that “Media” sample consisted of exosomes isolated from media alone with exosome free serum”. What do authors mean? I suppose tat they use exosome free serum but they were still able to precipitate exosome, as stated in the reply. However, this point in the text is still unclear. At line 100, authors wrote that “Exosomes were isolated from an equal volume of the cell culture media containing heat inactivated exosome-depleted fetal bovine serum media supplement (SystemBio Inc) has been which is referred to as “Media control”. if authors suspected/detected that exosome-free medium was not properly depleted, they should clearly explain that, as it is clearly explained in the reply.
Response: The media group consisted of cell culture media and exosome depleted serum. This group was used as a negative control. Exosomes were isolated from this group to verify if cell culture media is free from exogenous exosomes. We have made revisions in the methods section to clearly explain it (Line 99-101). We detected exosomes in “media” control because the exosome depleted serum was not completely free of exosomes. This was commercially acquired and possibly due to technical limitations there were some residual exosomes. We have modified the text in the results to indicate that the exosome free serum was not 100% free from exosomes. This modification is made in the results section (Line 190-193) and the discussion section (Line 299-303)
Comment 3: Line 192 refers to the same point and the sentence is again unclear “Although the cells were maintained in "Exosome free serum (SystemBio)" for the duration of the IL-1βß stimulation to avoid contamination with any serum exogenous exosomes, some residual exosomes were also detected in the exosome free media. This suggests that IL-1βß mediated astrocyte activation increases exosome secretion from primary astrocytes.” Why the presence of residual exosomes in the exosome free media suggests that IL-1βß mediated astrocyte activation increases exosome secretion from primary astrocytes? The Media alone sample should be clearly explained
Response: We appreciate the reviewer to highlight this sentence. We agree that it may be confusing for the readers. We have modified the sentence to clearly explain that IL-1β treatment increases exosome secretion from primary astrocytes (Line 188-190). In addition, media control which contained media along with exosome depleted serum also showed band for a light band for Alix marker suggesting that the exosome free serum contained some residual exosomes. The media along group is now clearly explained in the text (Line 190-193).
Comment 5: Line 96: Please correct the sentence “to pellet crude exosomal pellet”
Response: We have corrected the sentence as per the reviewer’s suggestion.
Comment 6: Line 274. The sentence “MRNAs for BCL-2, Bax and caspase 3 are among the few molecules which are involved in inflammatory and apoptotic pathway” is not precise, as many other molecules are involved. Authors should reformulate this sentence.
Response: We appreciate the reviewer’s comment regarding the clarity of this sentence. We have revised this sentence and have included additional details from the figure to increase the clarity.
Round 3
Reviewer 2 Report
The revised version of the manuscript has amended mistakes made in the 1st revision.